# Can New ENZIAN Score 2020 Represent a Staging System Improving MRI Structured Report?

**DOI:** 10.3390/ijerph18199949

**Published:** 2021-09-22

**Authors:** Lucia Manganaro, Veronica Celli, Miriam Dolciami, Roberta Ninkova, Giada Ercolani, Sandra Ciulla, Corrado De Vito, Stefania Maria Rizzo, Maria Grazia Porpora, Carlo Catalano

**Affiliations:** 1Department of Radiological, Oncological and Pathological Sciences, Sapienza University of Rome, Policlinico Umberto I, Viale del Policlinico 155, 00161 Rome, Italy; veronica.celli@uniroma1.it (V.C.); miriam.dolciami@uniroma1.it (M.D.); roberta.ninkova@gmail.com (R.N.); giada.ercolani@uniroma1.it (G.E.); sandra.ciulla@uniroma1.it (S.C.); carlo.catalano@uniroma1.it (C.C.); 2Department of Public Health and Infectious Diseases, Sapienza University of Rome, Policlinico Umberto I, Viale del Policlinico 155, 00161 Rome, Italy; corrado.devito@uniroma1.it; 3Faculty of Biomedical Sciences, University of Italian Switzerland (USI), Via Buffi 13, 6900 Lugano, Switzerland; StefaniaMariaRita.Rizzo@eoc.ch; 4Service of Radiology, Imaging Institute of Southern Switzerland, Clinica Di Radiologia EOC, 6900 Lugano, Switzerland; 5Department of Maternal and Child Health and Urological Sciences, Oncological and Pathological Sciences, Sapienza University of Rome, Policlinico Umberto I, Viale del Policlinico 155, 00161 Rome, Italy; mariagrazia.porpora@uniroma1.it

**Keywords:** ENZIAN score, magnetic resonance imaging, endometriosis, classification system, rASRM classification, pelvic pain

## Abstract

Structured reporting systems for endometriotic disease are gaining a central role in diagnostic imaging: our aim is to evaluate applicability and the feasibility of the recent ENZIAN score (2020) assessed by MRI. A total of 60 patients with suspected tubo–ovarian/deep endometriosis were retrospectively included in our study according to the following criteria: availability of MR examination; histopathological results from laparoscopic or surgical treatment; patients were not assuming estro-progestin or progestin therapy. Three different readers (radiologists with 2-, 5-, and 20-years of experience in pelvic imaging) have separately assigned a score according to the ENZIAN score (revised 2020) for all lesions detected by magnetic resonance imaging (MRI). Our study showed a high interobserver agreement and feasibility of the recent ENZIAN score applied to MRI; on the other hand, our experience highlighted some limitations mainly due to MRI’s inability to assess tubal patency and mobility, as required by the recent score (2020). In view of the limitations which arose from our study, we propose a modified MRI-ENZIAN score that provides a complete structured reporting system, more suitable for MRI. The high interobserver agreement of the recent ENZIAN score applied to MRI confirms its validity as a complete staging system for endometriosis, offering a shared language between radiologists and surgeons.

## 1. Introduction

### 1.1. Background

Endometriosis is a benign chronic multifocal inflammatory disease characterized by the growth of functional ectopic endometrial glands and stroma outside the uterus. It affects 10–15% of women in reproductive age and may cause non-cyclic chronic pelvic pain, dysmenorrhea, dyspareunia, urinary tract symptoms, and is frequently associated with infertility [1,2]. The peak of incidence is between 24 and 29 years old, and the clinical diagnosis of endometriosis is generally delayed by 6–7 years due to the broad spectrum of non-specific symptoms and psychosocial factors [3].

Laparoscopy with surgical biopsies is still considered the “gold standard” for the diagnosis of endometriosis, with histological verification of endometrial ectopic glands and/or stroma. Currently, two non-invasive modalities are routinely used for a presumptive diagnosis: transvaginal sonography (TVS) and magnetic resonance imaging (MRI). MRI has proved to be highly accurate in detecting localization of the endometriotic lesions and in obtaining accurate pre-operative mapping, particularly for deep infiltrating endometriosis (DIE), in order to plan a suitable surgical procedure [4,5]. Literature demonstrated the growing relevance of non-invasive procedures for the detection of endometriotic disease [6]. Actually, in accordance with several studies, during surgical procedures the presence of adhesions may limit the visualization of microscopic or small DIE implants during surgery, causing an incomplete removal of the lesions [7,8,9]. In contrast, the presence of adhesions does not cause a significant reduction in the diagnostic accuracy of MRI; this is particularly helpful in preoperative surgical planning where MRI avoids preoperative under-staging and under-treatment of patients with deep endometriosis [10]. Over the years, several laparoscopic and surgical scores for endometriosis have been compiled and then applied to preoperative MRI evaluation to attempt to unify language between surgeons and radiologists; nonetheless, currently, there is still wide debate about the staging of endometriosis.

The most widely used classification system throughout the world is the American Society for Reproductive Medicine classification score (ASRM), first presented by the American Fertility Society (AFS) in 1979 as AFS score, then revised in 1985 and again in 1997, when it was renamed the revised ASRM (rASRM) score [11]. This scoring system classifies endometriosis in four stages of increasing severity based mainly on the presence of endometriomas, peritoneal adhesions (ovarian/tubal) and obliteration of the pouch of Douglas. However, the rASRM score does not allow for a detailed description of the extent of DIE and neglects the involvement of the retroperitoneal organs [12].

In 2003 and 2005, the SEF (Stiftung Endometriose-Forschung; Scientific Endometriosis Foundation) published the ENZIAN classification (then revised in 2011), which provides a useful tool only for DIE classification in TVS, MRI, and surgical methods [13,14].

Hence, the latest revision of the ENZIAN score 2020 proposes a new comprehensive classification system, which includes the anatomical location and the size of all of the different forms of endometriotic lesions, the presence of adhesions, and the degree of involvement of the adjacent organs [15].

In fact, this scoring system combines a complete staging of deep endometriosis with the evaluation of peritoneal/ovarian/tubal localizations and the presence of adenomyosis. Therefore, it allows surgeons to plan a more complete and resolving surgical approach. 

### 1.2. Objective of the Study

In view of the fact that MRI may replace the preoperative laparoscopic examination in patients affected by endometriosis, our study aims to evaluate the applicability and reproducibility of the ENZIAN score on MRI and aims to highlight some limitations arising from the application of a laparoscopic score to radiological images.

## 2. Material and Methods

### 2.1. Population

Retrospective research conducted in our institute’s electronic database for the period between December 2020 and April 2021 revealed 60 patients with suspected tubo–ovarian/deep endometriosis whose data were considered for inclusion in our study. The study was approved by the local Ethics Committee of our Department. Patient age ranged from 22 to 49 years (mean age 33.5 years). We enrolled in our study patients who fulfilled the following criteria: (a) tubo–ovarian and/or deep endometriosis suspected at physical examination or TVS during gynecologic clinical assessment, (b) availability of MRI adequate protocol, (c) all pelvic MRIs included were performed at our institute, (d) availability of histopathological results from laparoscopic or laparotomic treatment, (e) preoperative assessment by pelvic MRI, and (e) patients were not assuming estro-progestin or progestin therapy. Exclusion criteria were as follows: (a) lack of available MR examination and/or (b) lack of definitive histopathological results.

### 2.2. MRI Technique

Pelvic MRI was performed in every case on the 3–T system GE Discovery 750. Patients were asked to follow a low-residue diet 3 days prior to MRI accompanied by enema with either rectal suppository pills (e.g., bisacodyl) or water the day before the study. Patients were instructed not to urinate for at least 1 h prior the MRI scan, in order to obtain a full bladder. Hyoscine N-butylbromide (Buscopan 20 mg/mL, Boehringer Ingelheim, 20 mg) was administered either by intravenous or muscular injection to reduce the normal bowel peristaltic artifacts. Patients were introduced into the gantry “feet first” in the supine position with one multichannel phased array surface body coil (32 channels, 127.73 MHz). The common study protocol included: single-shot fast-spin echo sequences (matrix 384 × 224, FOV 360 × 360, FA 90°, TR 2000, TE 102, and slice thickness 6 mm); T2 weighted FRFSE HR sequences (matrix 448 × 256, FOV 230 × 230, FA 90°, TR 6279, TE 1322 and slice thickness 3 mm); T1 w FSE sequences (matrix 320 × 192, FOV 240 × 240, FA 90°, TR 586, TE 8, and slice thickness 3mm); LAVA-flex sequences (matrix 288 × 224, FOV 310 × 310, FA 12°, TR 4, TE 2, and slice thickness 4 mm). MRI images were acquired according to multiple scan planes, in particular stacks were oriented on axial, coronal, and sagittal planes of the pelvis. Total acquisition time amounts to 23 min. 

### 2.3. Image Analysis

Endometriotic disease was evaluated separately by three different readers: two young radiologists with two- and five-years’ experience in pelvic MRI, respectively, and a senior radiologist with 20 years of experience in pelvic MRI, with a particular interest and expertise in endometriosis. Two radiologists (S.C. and G.E.) selected the patients who fulfilled the previously mentioned inclusion criteria; then, three readers (L.M., V.C., and M.D.) independently reviewed the images, blind to clinical history and examination, or histopathological results.

Radiologists were required to assign a score according to the revised ENZIAN score 2020 [15] and to compile a structured template following parameters reported below, as is summarized in Figure 1 and Figure 2:

*Peritoneum (P):* considers all superficial (sub-peritoneal invasion <5 mm) peritoneal implants, descripted as follows: P1 < 3 cm (sum of all maximal diameter); P2 = 3–7 cm (sum of all maximal diameter); P3 > 7 cm (sum of all maximal diameter). 

*Ovary (O):* considers all endometriomas and infiltrating ovarian surface foci (≥5 mm) separately calculated for each side (annotated as “l” = left and “r” = right). The description is as follows: O1 < 3 cm (sum of all maximal diameter); O2 = 3–7 cm (sum of all maximal diameter); O3 >7 cm (sum of all maximal diameter). 

*Tubo–ovarian* condition *(T):* presence of adhesions between the ovary and pelvic sidewall +/- tubo–ovarian adhesions (T1); T1 *plus* adhesions to the uterus *or* isolated adhesions between the adnexa and uterus (T2); T2 *plus* adhesions to the USL and/or bowel *or* isolated adhesions between the adnexa and the uterosacral ligaments (USLs)and/or bowel (T3). The recent ENZIAN score 2020 would also include information regarding mobility of the ovaries and tubes, and tuba patency, but the readers are not required to answer these fields because they are poorly evaluated by MRI.

*Deep infiltrating endometriosis (DIE):* considers all implants showing sub-peritoneal infiltration of >5 mm. The ENZIAN score defines the lesions’ base on the site and the different organs involved; they are classified in Compartment A represented by vagina or recto–vaginal space or retrocervical area *(measured on cranio–caudal axis = sagittal plane),* Compartment B represented by uterosacral and cardinal ligaments or pelvic sidewall *(measured on mediolateral axis = axial plane),* and Compartment C represented by rectal or colic wall up to 16 cm from the anal verge *(measured on cranio–caudal axis = sagittal plane)*. Lesions are also defined by size as follows: A/B/C 1 < 1 cm (sum of all maximal diameter); A/B/C 2 = 1–3 cm (sum of all maximal diameter); A/B/C 3 > 3 cm (sum of all maximal diameter). Each compartment is descripted separately.

*Adenomyosis and other extragenital deep endometriosis*: the presence of uterine adenomyosis, defined as thickening of the myometrium–endometrium junction line greater than 12 mm (FA); bladder lesions with involvement of muscular layer (FB); ureteral lesions with involvement of muscular layer (extrinsic and/or intrinsic; FU); sigmoid colon/coecum/ileum lesions above 16 cm from the anus (FI); other lesions such as diaphragms, liver, abdominal wall, etc. annotated as F (diaphragms), F (liver), or F (abdominal wall). Paired organs (ovary, tube, USLs) are annotated accordingly to the side, a slash separates left/right ( / ), both sides are annotated, even if only one side is affected; the annotation of **m** = missing organ and **x** = not visualized or unknown is used for ovaries and tubes. 

For each patient, each radiologist separately assigned a score structured as follows:

**ENZIAN** **P**_, **O**_/_, **T _/_**, **A _**, **B**_/_, **C** _, **F** _( )...... 

A number or letter (m/x) representative of the score is then assigned to each compartment and placed after the capital letter; number 0 is used in case of non-involvement.

## 3. Result

By comparing the MRI-ENZIAN score assigned by each radiologist, K Coen was 0,73074; concordance was excellent for peritoneal implants (0.8912), for adnexal lesions (0.8153) and for uterine adenomyosis (1.000); a good concordance was also evaluated for vagina–rectovaginal space (0.7645), for USLs (0.7402), for rectum (0.7932), and extragenital localization (0.6349). The concordance was poor only for lesion involving the tubo–ovarian complex (0.5455) (Table 1).

## 4. Discussion

Endometriosis disease is frequently undiagnosed or under-staged and this condition is particularly significant for DIE lesions [16,17]. Additionally, the visualization of DIE is often difficult and incomplete through a surgeon’s approach because of the presence of the fibrotic adhesions which often coexist with DIE.

Several studies reported that MRI evaluation is highly accurate for preoperative surgical planning [10], in accordance with the European Society of Urogenital Radiology’s (ESUR) guidelines 2017 [6]; in fact many large centers utilize these studies routinely, especially in the case of equivocal recto–vaginal, ureteral, or bladder endometriosis [18,19,20]. It follows that MRI is assuming an increasingly central role in the pre-operative diagnosis of endometriotic disease [21,22,23,24,25].

For this purpose, it is particularly important to establish a standardized staging system to communicate with surgeon, and to improve the preoperative staging and the treatment of deep endometriosis [26]. Over the years, several surgical scores have been applied to radiological reports in order to identify a shared language between radiologists and surgeons.

The widely accepted rASRM has certain limitations because of its incomplete description of DIE. Comparing rASRM and previous ENZIAN score (2011), Montanari et al. highlighted these limitations by assessing a significant association between DIE extent and symptoms when described by the ENZIAN classification (2011) [27]. Furthermore, the ENZIAN score 2011 showed a good correlation between preoperative MRI features and intraoperative findings in patients with DIE [24,28]. Recent literature has also evaluated inter-reader agreement with this classification, showing varying results; Thomassin-Naggara et al.’s trial (2020) consisted of 150 cases affected by DIE. They found excellent inter-reader agreement for A and C compartments but poor agreement for the B compartment, in accordance with previous publications by Saba et al. (2010) and Bazot et al. (2011) [29,30,31]. On the other hand, the recent trial conducted by Burla et al. (2020), which consisted of 21 cases of DIE, showed an overall lower concordance, with a particularly weak inter-rater agreement for compartment C [32].

Nonetheless, even the ENZIAN score 2011 provides an incomplete description because it does not include peritoneal implants, ovarian locations or adhesions.

Recently, a further improvement has been achieved by the newly proposed ENZIAN score 2020; it provides a complete staging system evaluating in detail the different forms of endometriosis (peritoneal implants, endometriomas, adhesions, and deep endometriosis) that can be applied by surgeons, sonographers, and radiologists following the same principles.

An additional advantage of the ENZIAN score consists of the application of a structured reporting which, over the last few decades, has gained relevance in radiological research due to its straightforward approach. The use of a structured reporting system has already been studied with positive results for adnexal, pancreatic, prostatic, colonic, anorectal, and hepatic diseases, succeeding in providing maximum value for personalized patient care [33,34,35]. The first application of structured reporting in gynecologic pelvic disease was introduced by Franconieri et al. (2018) regarding uterine fibroids [36]. Moreover, some studies have shown that structured reporting is associated with greater sensitivity in identifying endometriotic disease, because it includes all relevant information for procedural planning, which is frequently missed in narrative reports [18,37].

Increasing the complementarity of practices between radiologists and gynecologic surgeons has the potential to improve surgical decision making; this is particularly important in those cases where surgery is the only resolving treatment.

Jörg Keckstein et al. have recently affirmed that the newly prosed ENZIAN score should be applied by surgeons, sonographers, and radiologists following the same principles, but their study only provides very detailed guidance for its application in case of TVS [15].

Therefore, our study aims to validate the applicability and reproducibility of the recent ENZIAN score to MRI and proposes some modifications that would render the score more suitable for MRI evaluation. Actually, our study confirmed that the ENZIAN score represents a useful overall applicable staging system for MRI. In particular, our results showed excellent interobserver agreement for peritoneal implants (0.8912), for adnexal lesions (0.8153), and for uterine adenomyosis (1.000); moderate concordance for deep endometriosis implants (K Coen vagina–rectovaginal space = 0.7645, K Coen uterosacral ligaments (USLs) = 0.7402, K Coen rectum = 0.7932), which emphasis that the correct assessment of DIE is more influenced by the reader’s expertise in endometriosis. The principal limitation of the score is represented by the evaluation of tubo–ovarian complex (T); in fact, our experience highlighted both poor interobserver agreement (K Coen = 0.5455) and poor applicability to MRI of some parameters regarding the tubo–ovarian condition (T).

In fact, even if MRI is able to evaluate the presence of adhesion between the tubo–ovarian complex and the surrounding structures (pelvic sidewall, uterus, USL, bowel), MRI is not able to assess the mobility of the tubo–ovarian complex and the tubal patency. These represent the main differences compared to TVS evaluation. In fact, Jörg Keckstein et al. were able to assess the reduction or loss of mobility between these structures, separately for the left and right side; this assessment was carried out by TVS, through the evaluation of the sliding sign of the named structures. Moreover, tubal patency can optionally be assessed using hysterosalpingo contrast sonography.

On the contrary, direct assessment of tubal patency is not possible through static MRI. Nonetheless, this condition could be indirectly assessed by the evidence of sactosalpinx. In fact, sactosalpinx represents a fluid-filled dilatation of the fallopian tube which occurs when the tube is occluded at its distal end or both ends; however, this condition is not currently considered by the recent ENZIAN score.

In our experience the following parameters should be included in the MRI ENZIAN score to improve the diagnostic accuracy of tubo–ovarian complex:-Evaluate presence or absent of sactosalpinx, annotated with “+” (presence) or “-” (absence);-Specify which type of fluid is contained, annotated with H = hematic (hematosalpinx), or F = simple fluid (hydrosalpinx), or C = corpuscular (probable pyosalpinx);-Indicate the largest diameter measured in the point of greatest distension of the tube.

In addition, it would be useful to introduce in the structured reporting the measurement of all DIE implants, both for pelvic and extra-pelvic sites (FB, FI, FU, F diaphragm/lung/nerve), using the same system applied for compartments A, B, and C, and describing it as follows: F(B/I/U) 1 < 1 cm, F(B/I/U)2 = 1–3 cm, F(B/I/U) 3 > 3 cm.

Moreover, the ongoing dialogue with surgeons has highlighted the need to provide a more detailed description of bladder and ureteral localizations. Currently, a wide choice of surgical approaches is available (endoscopic management, ureterolysis, ureteral resection with ureteral reconstruction, and ureteroneocystostomy), and preoperative imaging is required for operative management planning. Surgery depends on the type, extent, and location of the disease, and MRI allows for the assessment of the extent of the disease in the bladder and ureters [38]. Through MRI, it is also possible to distinguish the two main types of ureteric endometriosis based on the grade of infiltration of ureteral wall: the intrinsic form and the extrinsic forms.

In view of the surgical parameters, we believe it is appropriate to include in the structured report:
-The measurement of the distance between the endometriotic implants and the vesicoureteral–uterine junction measured in cm, both for ureteral and bladder localizations, descripted as follows: “VUJ (l)) e.g., 3 cm” and VUJ (r)) e.g., 3 cm” where “r” and “l” indicate, respectively, the left and right vesicoureteral junction;-The difference between intrinsic (annotated with “i”) from extrinsic form (annotated with “e”) of ureteral endometriosis.

## 5. Conclusions

Our experience highlights that the new ENZIAN score shows high concordance in results for endometriosis evaluation, despite readers’ differing experiences in female pelvis imaging (two young radiologists vs. a senior radiologist), because it provides a systematic roadmap for objective evaluation of the location and the quantification of endometriosis disease; therefore, it allows the assessment of the degree of severity of endometriotic disease. 

Additionally, the applicability of this staging system would help to standardize language among radiologists, and between radiologists and surgeons, by including all information relevant to procedural planning in the structured report.

## Figures and Tables

**Figure 1 ijerph-18-09949-f001:**
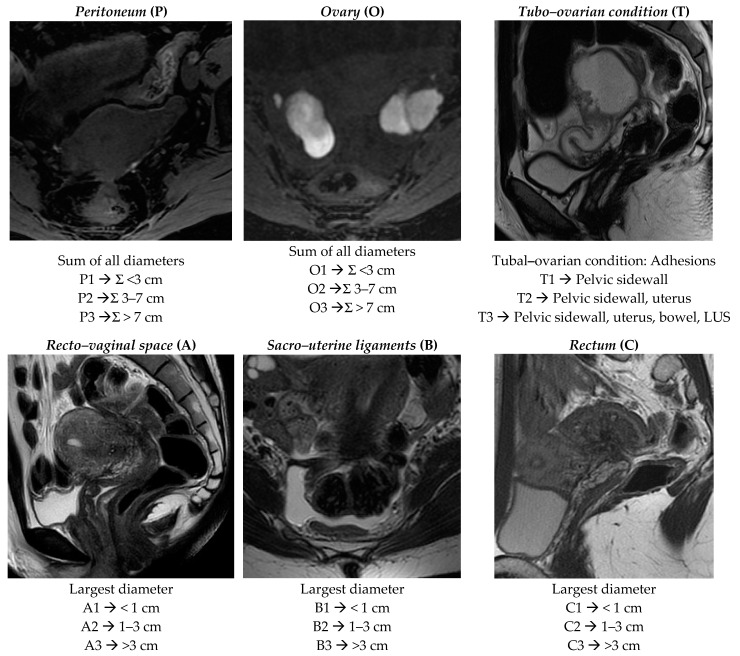
ENZIAN score 2020: MRI overview of endometriotic pelvic implants affecting different organs and compartments.

**Figure 2 ijerph-18-09949-f002:**
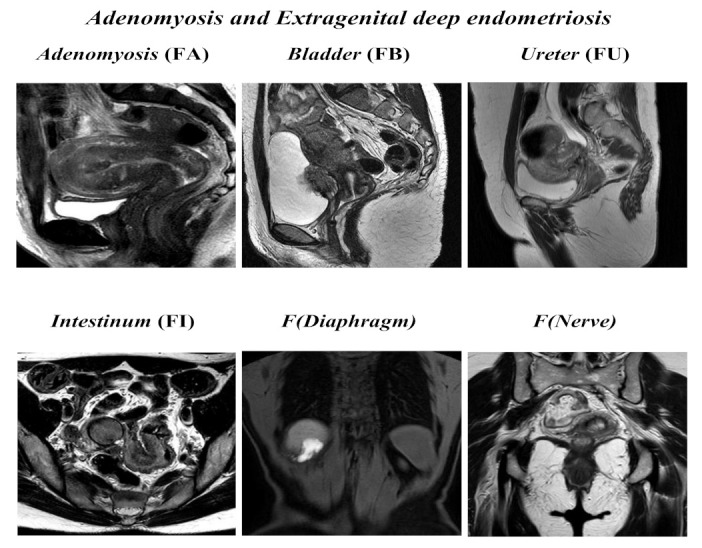
ENZIAN score 2020: MRI overview of extragenital deep endometriosis and adenomyosis.

**Table 1 ijerph-18-09949-t001:** ENZIAN staging for all observed lesions with relative degrees of concordance between readers (k-Cohen).

SITE	Kappa	Z
*Peritoneum* (P)	0.8912	17.28
*Ovary* (O)	0.8153	17.20
*Tubo–ovarian condition* (T)	0.5455	10.63
*Recto**–vaginal space* (A)	0.7645	12.50
*Sacro**–uterine/cardinal ligaments* (B)	0.7402	15.70
*Rectum* (C)	0.7932	15.91
*Extragenital deep endometriosis* (F)	0.6349	12.48

“SITE” represents different pelvic regions considered separately by each reader applying the score. “K” represents degree of concordance between readers expressed through the following reading criteria: k < 0 non concordance; k 0–0.4 poor concordance; k 0.4–0.6, fair concordance; k 0.6–0.8, good concordance; k 0.8–1, excellent concordance.

## Data Availability

The data presented in this study are available on request from the corresponding authors.

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
