# Peer review of "Can New ENZIAN Score 2020 Represent a Staging System Improving MRI Structured Report?"

_ijerph, 2021, doi:10.3390/ijerph18199949_

Round 1

Reviewer 1 Report

The topic of the work is interesting and up-to-date; raises an important issue which is the diagnosis of endometriosis.

The following points should be addressed before publication

  1. The aim of the work is understandable, but it should be extracted from the text in a separate paragraph.

  1. In my opinion, conclusions should be extracted from the text.

  1. The material and methods contain information that is not entirely clear; the inclusion criteria were:
  • tubo-ovarian and/or deep endometriosis suspected at physical examination or TVS,….

            I do not know if I understand correctly, but it follows that radiologists assessed the images         without a preliminary diagnosis of suspected endometriosis?

…They independently reviewed the images, blind to clinical history and examination or histopathological results…

            In my opinion, the authors should explain who included patients in the study (radiologist or gynecologist or…?) and qualified them or their MRI images for assessment by three independent radiologists.

line 44. it should be space before [ ]; the same in line ; 52;  67; 190; 212

line 45:  [3].

line 74: [13,14]

line 53: for instead of per

line 83-84: replace the preoperative laparoscopic examination- did the authors mean replacing laparoscopy with MRI?

line 100: laparoscopic or surgical- was it to distinguish between laparoscopy and laparotomy?

line 128-139:  isolated adhesions Author contributions between the adnexa and uterus

line 288: vescicoureteral    vesicoureteral

line 383: #Enzian  ; to remove  #

line 52: recent ….reference from 2017; it is not recent study

line 54: Actually, in according to recent literature… references [7,8,9] from 2015; 2009; 2012 are not recent study; “recent”  should be replaced another word (in some study etc.) or references should be replaced with newer items

line 229: similar situation; reference [37] from 2015

Author Response

Thank you for your appreciated revisions and corrections.

Responses to your comments are listed below.

Point 1:“The aim of the work is understandable, but it should be extracted from the text in a separate paragraph”

Response 1: Thank you for your comments, we accepted the revision, and we extracted the aim from the text in a separate paragraph.

Point 2: In my opinion, conclusions should be extracted from the text.

Response 2: Thank you for your comments, we accepted the revision, and we extracted the aim from the text in a separate paragraph.

Point 3.1:  tubo-ovarian and/or deep endometriosis suspected at physical examination or TVS,….  “ I do not know if I understand correctly, but it follows that radiologists assessed the images   without a preliminary diagnosis of suspected endometriosis?   

Response 3.1: Thank you for your comments, we accepted the revision and modified the inclusion criteria.

Point 3.2: “They independently reviewed the images, blind to clinical history and examination or histopathological results…” In my opinion, the authors should explain who included patients in the study (radiologist or gynecologist or…?) and qualified them or their MRI images for assessment by three independent radiologists.

Response 3.2: We accepted the revision and clarified who included the patients modifying the text.

Point 4: Line 44. it should be space before [ ]; the same in line ; 52;  67; 190; 212

Response 4: We accepted the revision and modified the text.

Point 5: line 45:  [3].

Response 5: We accepted the revision and modified the text.

Point 6: line 74: [13,14]

Response 6: We accepted the revision and modified the text.

Point 7: line 53: for instead of per

Response 7: We accepted the revision and modified the text.

Point 8: line 83-84: replace the preoperative laparoscopic examination- did the authors mean replacing laparoscopy with MRI?

Response 8: Yes, we confirmed that MRI may replace laparoscopy for the pre-operative examination.

Point 9: Line 100: : laparoscopic or surgical- was it to distinguish between laparoscopy and laparotomy?

Response 9: Yes, we wanted to distinguish between laparoscopy and laparotomy; we accepted the revision and replaced ”surgical” with “laparotomic”.

Point 10:  line 128-139:  isolated adhesions Author contributions between the adnexa and uterus

Response 10: We accepted the revision and eliminated “Author contribution” from the text.

Point 11: line 288: vescicoureteral    vesicoureteral

Response 11: We accepted the revision and replaced “vescicoureteral ” with  vesicoureteral.

Point 12: line 383: #Enzian  ; to remove  #

Response 12: We accepted the revision and eliminated  #.

Point 13: line 52: recent ….reference from 2017; it is not recent study

Response 13: We accepted the revision and replaced “recent” with “literature”.

Point 14: line 54: Actually, in according to recent literature… references [7,8,9] from 2015; 2009; 2012 are not recent study; “recent”  should be replaced another word (in some study etc.) or references should be replaced with newer items

Response 14: We accepted the revision and replaced “recent literature” with “several studies”.

Point 15: line 229: similar situation; reference [37] from 2015

Response 15: We accepted the revision and replaced “recent” with “moreover”.

Reviewer 2 Report

The manuscript shows overview of MRI of endometriosis with ENZIAN score. 

  1. Whether this new scoring system improves MRI in other disease conditions such as cancer as well as in different organs?
  2. Result part must be elaborated.                                                                                                                                                                                                 I don't find any supplementary files.

Author Response

Response to Reviewer 2 Comments:

Thank you for your appreciated revisions and corrections.

Responses to your comments are listed below.

Point 1: Whether this new scoring system improves MRI in other disease conditions such as cancer as well as in different organs?

Response 1: It is certainly very interesting and useful to use a scoring system also for other pelvic pathologies; however, the Enzian score includes all the relevant informations for gynecologists to plan an optimal surgical planning for endometriotic disease. It would be appropriate to establish a dedicated score for each pelvic disease after a comprehensive interdisciplinary discussion, as occurred with ADNEX score for the classification of ovarian masses on MRI.

Point 2: Result part must be elaborated

Response  2:Our study aims to improve the current scoring system highlighting its advantages and limitations. We did this through the statistical analysis of the concordance of the readers' results, but mainly through the "qualitative analysis" of the scoring which is fully explained in the discussion. So our primary goal was not to find a statistical correlation to be expressed in the results but rather to give a commentary to improve the score, as it is expressed in the discussion; for these reasons we have privileged and deepened more the part of the discussions than the results.

Reviewer 3 Report

Dear Authors,

your article is interesting in my opinion.

I have some small suggestions:

  1. Think about the legend under Table 1
  2. Some small english corrections are needed
  3. Is ellipsis in line 170 essential?

Author Response

Response to Reviewer 3 Comments:

Thank you for your appreciated revisions and corrections.

Responses to your comments are listed below.

Point 1:Think about the legend under Table 1

Response 1:We accepted the revision and clarified the legend as follows: 

ENZIAN staging for all observed lesions with relative degree of concordance between readers (k-Cohen): "SITE" represents different pelvic regions considered separately by each reader applying the score. “K” represents degree of concordance between readers expressed through the following reading criteria: k<0 non concordance; k 0-0,4, poor concordance; k 0,4-0,6, fair concordance; k 0,6-0,8, good concordance ; k 0,8-1, excellent concordance.

Point 2: Some small english corrections are needed

Response 2: We accepted the revision and modified the text.

Point 3: Is ellipsis in line 170 essential?

Response 3:The “( )”after “F” suggests that for “Extragenital deep endometriosis” the reader should clarify which structure is affected and fulfil the parenthesis with : (FB) if some bladder lesion ,(FU) if some ureteral lesions,  (FI) if some sigma/coecum/ ileum lesions above 16 cm from the anus; F (diaphragms), F (liver), F (nerve) if some diaphragmatic hepatic or nerve implants are presents.

Reviewer 4 Report

Reviewer Notes

Line 42: “no-cyclic chronic pelvic pain” implies that it does not produce cyclic pelvic pain.  Suggest “non-cyclic”

Line 45:  You comment on the delay in diagnosis, but it is not clear what the relevance or context around this statement is.  Is this the delay in clinical, radiologic, surgical or histologic diagnosis?  Does this imply that patients do not receive any therapy in symptoms over this time or that they do not have surgery during this period only?  Please claify.

Line 45: Citation 3 is given following punctuation here.  Elsewhere citations are given prior to punctuation.  Please be consistent throughout the manuscript.

Line 51: Remove “an” from “obtaining an accurate mapping”

Line 54: “in according to” is not syntactically correct.

Line 56: No studies to date have demonstrated a cause and effect relationship between not excising “deep microscopic implants” and recurrence of pain/symptoms.  In fact many who have complete excisional procedures of early stage disease still have recurrence of pain/symptoms.  It is inappropriate to imply cause and effect when citing these studies when they were not designed to show this.

Line 59: “undertreatment” is two words

Line 60: “Laparoscopic” and “Surgical” are not proper nouns

Line 71: “organs” does not need to be pleural

Line 73: “revisited” does not make sense in this sentence

Line 74: “both” implies two list items but your list has three

Line 75: The referencing uses sets of square brackets for each reference which is different than elsewhere.  Please be consistent throughout the manuscript

Line 82: What is a “resolving surgical approach”?

Line 92: “A retrospective research” is no worded properly

Line 97: “Patients” is not a proper noun

Line 105: “cases” need not be pluralized

Line 107: Why did patients require bowel preparation prior to MRI?  Is there evidence that this improves detection rates of DIE?

Line 123: The font size changes here

Line 139: “Author”?

Line 139: USL is an undeclared acronym.  It is only declared on line 178.

Line 142: Spelling mistake: “tuba”

Line 153: Spelling mistake: “descripted”

Line 157: “sigma”?

Line 165: LUS is an undeclared acronym.

Line 169: “Patient” should not be capitalized

Line 171-3: This sentence is hard to read and should be reworded

Line 179: “lesion” should be pleural

Discussion: the language used does not flow well and has syntactical errors and should be reworded. “European society of urogenital radiology” should have each word capitalized. Instead of advocating for MRI over exploratory laparoscopy, I suggest a different perspective to take.  Most experts agree that exploratory laparoscopy performed for the purpose of diagnosis should NEVER be done.  Laparoscopy can be performed as a see and treat model or as a planned complete procedure with a surgeon or surgical team with the expertise to do so.  The argument to use MRI should be done so without comparing it to a practice that should never be done, but rather some practice with value.

Author Response

Response to Reviewer 4 Comments:

Thank you for your appreciated revisions and corrections.

Responses to your comments are listed below.

Point 1 : Line 42: “no-cyclic chronic pelvic pain” implies that it does not produce cyclic pelvic pain.  Suggest “non-cyclic”

Response 1: We accepted the revision and modified the text.

Point 2: Line 45:  You comment on the delay in diagnosis, but it is not clear what the relevance or context around this statement is.  Is this the delay in clinical, radiologic, surgical or histologic diagnosis?  Does this imply that patients do not receive any therapy in symptoms over this time or that they do not have surgery during this period only?  Please clarify.

Response 2: Thank you for your comment, according to literature, the wide spectrum of non-specific symptoms and psychosocial factors associated with endometriosis cause a delay (estimated between 7-8 years) in clinical suspicion of endometriosis; these lead to a delay in diagnosis and proper treatment, both surgical and pharmacological.

We have clarified this part in the text. 

(Ghai, V., Jan, H., Shakir, F., Haines, P., & Kent, A. Diagnostic delay for superficial and deep endometriosis in the United Kingdom. Journal of obstetrics and gynaecology : the journal of the Institute of Obstetrics and Gynaecology 2020, 40(1), 83–89. https://doi.org/10.1080/01443615.2019.1603217

Simpson, C. N., Lomiguen, C. M., & Chin, J. Combating Diagnostic Delay of Endometriosis in Adolescents via Educational Awareness: A Systematic Review. Cureus 202013(5), e15143. https://doi.org/10.7759/cureus.15143 

Surrey, E., Soliman, A. M., Trenz, H., Blauer-Peterson, C., & Sluis, A. Impact of Endometriosis Diagnostic Delays on Healthcare Resource Utilization and Costs. Advances in therapy 202037(3), 1087–1099. https://doi.org/10.1007/s12325-019-01215-x ).

Point 3: Line 45: Citation 3 is given following punctuation here.  Elsewhere citations are given prior to punctuation.  Please be consistent throughout the manuscript.

Response 3: We accepted the revision and modified the text.

Point 4 : Line 51: Remove “an” from “obtaining an accurate mapping”

Response 4: We accepted the revision and modified the text.

Point 5: Line 54: “in according to” is not syntactically correct.

Response 5: We accepted the revision and modified the text.

Point 6:Line 56: No studies to date have demonstrated a cause and effect relationship between not excising “deep microscopic implants” and recurrence of pain/symptoms.  In fact many who have complete excisional procedures of early stage disease still have recurrence of pain/symptoms.  It is inappropriate to imply cause and effect when citing these studies when they were not designed to show this.

Response 6: Thank you for your comment. We agree that recurrences can occur also after a complete excision of the

lesions. The concept that we wanted to explain was that adhesions can cause a difficult detection of lesions

at laparoscopy and therefore an incomplete removal of them that can be wrongly considered as a recurrence.

Therefore, preoperative MRI which is not influenced by the presence of adhesions should be recommended

to guide the surgeon to perform a complete resection of the lesions.

We have rephrased the text as follows:   

“the presence of adhesions may limit the visualization of microscopic or small DIE implants during surgery

causing an incomplete removal of the lesions (7–9)”.

Point 7: Line 59: “undertreatment” is two words

Response 7: We accepted the revision and modified the text.

Point 8: Line 60: “Laparoscopic” and “Surgical” are not proper nouns

Response 8: We accepted the revision and modified the text.

Point 9: Line 71: “organs” does not need to be pleural

Response 9:We accepted the revision and modified the text.

Point 10:Line 73: “revisited” does not make sense in this sentence

Response 10: We accepted the revision and modified the text.

Point 11: Line 74: “both” implies two list items but your list has three

Response 11: We accepted the revision and modified the text.

Point 12: Line 75: The referencing uses sets of square brackets for each reference which is different than elsewhere.  Please be consistent throughout the manuscript

Response 12: We accepted the revision and modified the text.

Point 13: Line 82: What is a “resolving surgical approach”?

Response 13: Thank you for your question, we meant a surgical approach that removes all endometriotic implants, but because we talked about surgical planning we decided to modify the text.

Point 14: Line 92: “A retrospective research” is no worded properly

Response 14: We accepted the revision and modified the text.

Point 15: Line 97: “Patients” is not a proper noun

Response 15: We accepted the revision and modified the text.

Point 16: Line 105: “cases” need not be pluralized

Response 16: We accepted the revision and modified the text.

Point 17:Line 107: Why did patients require bowel preparation prior to MRI?  Is there evidence that this improves detection rates of DIE?

Response 17: According to ESUR guideline, bowel preparation is advocated as ‘best practice’ for the detection of DIE.

Point 18: Line 123: The font size changes here

Response 18: We accepted the revision and modified the text.

Point 19: Line 139: “Author”?

Response 19: We accepted the revision and modified the text.

Point 20: Line 139: USL is an undeclared acronym.  It is only declared on line 178.

Response 20: We accepted the revision and modified the text.

Point 21: Line 142: Spelling mistake: “tuba”

Response 21: We accepted the revision and modified the text.

Point 22: Line 153: Spelling mistake: “descripted”

Response 22: We accepted the revision and modified the text.

Point 23: Line 157: “sigma”?

Response 23: We accepted the revision and modified the text.

Point 24: Line 165: LUS is an undeclared acronym.

Response 24: We accepted the revision and modified the text.

Point 25: Line 169: “Patient” should not be capitalized

Response 25: We accepted the revision and modified the text.

Point 26: Line 171-3: This sentence is hard to read and should be reworded

Response 26: We accepted the revision and modified the text as follows:

“A number or letter (m/x) representative of the score is then assigned to each compartment and placed after the capitol letter; number 0 is used in case of non-involvement.”

Point 27: Line 179: “lesion” should be pleural

Response 27: We accepted the revision and modified the text.

Point 28: Discussion: “European society of urogenital radiology” should have each word capitalized.

Response 28: We accepted the revision and modified the text.

Point 29: Discussion: Instead of advocating for MRI over exploratory laparoscopy, I suggest a different perspective to take.  Most experts agree that exploratory laparoscopy performed for the purpose of diagnosis should NEVER be done.  Laparoscopy can be performed as a see and treat model or as a planned complete procedure with a surgeon or surgical team with the expertise to do so.  The argument to use MRI should be done so without comparing it to a practice that should never be done, but rather some practice with value.

Response 29: Thank you for your appreciated comment, we agreed with you and the current view that the use of diagnostic laparoscopy as a preoperative investigation is outdated. We have rephrased the sentence and removed the comparison to laparoscopy.